# Maternal and Infant Health in Abu Dhabi: Insights from Key Informant Interviews

**DOI:** 10.3390/ijerph16173053

**Published:** 2019-08-22

**Authors:** Hazel Gardner, Katherine Green, Andrew Gardner, Donna Geddes

**Affiliations:** 1School of Molecular Sciences, University of Western Australia, Crawley 6009, Australia; 2School of Education, Capella University, 225 South 6th St, Minneapolis, MN 55402, USA

**Keywords:** maternal health, infant health, Abu Dhabi, United Arab Emirates, key informants

## Abstract

Consequent upon rapid development in Abu Dhabi, there has been a rise in chronic disease, the susceptibilities to which are influenced by events occurring in early life. Hence, maternal and infant health are key areas in public health policy. Following a study of maternal and infant health in a cohort of mothers in Abu Dhabi between 2002 and 2004, seven key informant interviews were undertaken to elucidate the study findings through the impressions of Emirati women in positions within the healthcare area—including ministries, hospitals, and universities in Abu Dhabi. Semi-structured interviews were based on five key questions that covered the cultural responsiveness of the maternal health services—breastfeeding, health education, and physical and recreational activity. The responses were analysed using a thematic content technique and indicated that the status of women, cultural beliefs and practices, limited health knowledge, and language differences between the local population, healthcare providers, and health promoting materials were important themes. The study highlighted areas for future research and policy, including the communication gaps between healthcare professionals and women, the influences of advertising and the media on health issues, heath education, and ways to increase women’s participation in physical exercise. It is vital to consider non-medical determinants of health alongside biomedical determinants, to help develop culturally appropriate health strategies for this population.

## 1. Introduction

The United Arab Emirates (UAE) is a country undergoing rapid development in both society and infrastructure. The transition from traditional pastoral and fishing communities with rudimentary health services, education or material wealth, to a modern urban lifestyle has occurred mainly over the last fifty years or two generations. Consequent upon this rapid change has been a rise in chronic disease; particularly obesity, heart disease and diabetes [1]. Susceptibility to chronic disease is influenced by events occurring in early life [2,3]. Hence, maternal and infant health are key areas in public health policy.

The UAE has an unusual demography, with expatriates and migrant workers constituting approximately 81% of the Abu Dhabi population [4]. Due to this influx of migrant workers, more than 63.9% of the population are male. Hence, the public health system of the country must cater for a diverse population in addition to its own Emirati citizens. The population of Emirati citizens has particular requirements, including higher proportions of women, children and the elderly [4], together with their own particular cultural background.

This paper reports the responses of a group of key informants who took part in semi-structured interviews to enhance understanding of the findings of a study of Emirati maternal and infant health in the emirate of Abu Dhabi (hereafter referred to as the cohort study) [5,6,7], and to explore policy and practice implications and recommendations arising from this study. The key results from the cohort study are captured in the questions put to the key informants.

A good key informant is seen by Merriam as someone who: “understands the culture but is also able to reflect on it and articulate for the researcher what is going on” [8]. The key informants who took part in this study were all Emirati women with roles in the development of maternal and infant health policy in the emirate of Abu Dhabi. They therefore possessed in-depth knowledge of Emirati maternal and infant health, and could offer valuable perspectives. Each had special knowledge, skills or status [9] that they were willing to share to enhance understanding and the interpretation of the cohort study’s results. Key informants were therefore not selected by random sampling, but through purposive sampling based on their potential contribution to enhancing the understanding of Emirati maternal and infant health in the Emirate of Abu Dhabi.

The objective of conducting key informant interviews was to elucidate the results of the cohort study of mothers and infants, and to use their insights to formulate recommendations that might improve maternal and infant health outcomes in the Emirate of Abu Dhabi. The interviews were semi-structured, guided by carefully selected research questions. The use of a semi-structured interview format ensured that comparable data would be collected across participants to facilitate analysis [10]. The use of individual rather than group interviews ensured that the views of each key informant were recorded without being influenced by the opinions of others [11].

## 2. Methods

The interviews were held either at the informant’s own professional office or at Zayed University. All key informants were competent English speakers, negating the need for translation. Informed consent was received from all the participants, who were provided with background information on the cohort study, information relating to the key findings from the cohort study and the selected questions. The interviewer (HG) had lived in Abu Dhabi for seven years and during that time had developed an extensive understanding of the culture, working directly with Emirati girls, young women and mothers. As the interaction between the interviewer and key informant is crucial to the success of the interview, it was important that the interviewer developed a personal rapport with each key informant, so that a sense of trust was generated [12]. The interviewer was known to the informants in a professional context, and Zayed University was considered locally to provide high quality education to Emirati women. All the key informants had a strong interest in maternal and infant health and welcomed the opportunity to contribute to the development of knowledge and recommendations on health issues affecting Emirati women and infants in the UAE.

### 2.1. Selection of Factors Influencing Maternal and Infant Health 

Selected factors influencing maternal and infant health identified from the cohort study are presented in Table 1.

### 2.2. Interview Questions

Interview questions were developed, based on the factors in Table 1, as follows:After reviewing the key findings from the study on Emirati maternal and child health, what do these results mean to you? What other issues might you consider to be equally problematic?How could maternal health services be more culturally responsive to the needs of Emirati women?What measures might be taken to encourage Emirati women who are working or studying to continue breastfeeding?Given the finding that many of the participants were not well-informed regarding health issues, how might health education be enhanced particularly for Emirati women with lower levels of education (e.g., relating to sudden infant death syndrome SIDS, anaemia, obesity, folic acid, infant feeding, or healthy eating)?How can Emirati women be encouraged to participate in more physical or recreational activity?

These five questions provided a guide for the interviews, but were broad enough to allow key informants to express opinions and to offer additional comments.

### 2.3. The Key Informants

The key informants were seven Emirati women drawn from a variety of backgrounds, all with involvement in Emirati maternal and infant health services. They are listed in Table 2. 

### 2.4. Analysis of the Key Informant Interviews

The responses were recorded by hand, as women in Emirati culture often prefer not to be recorded either in audio or visual formats. Analysis of the key informant interviews was undertaken using the thematic content analysis technique described by Burnard [14] and described in the Appendix A. This method is based on the work of Glaser and Strauss [15] and other researchers who have used content analysis and qualitative data analysis extensively [16,17,18,19,20,21]. Thematic content analysis was well suited to this study as it assumes that the interview structure is open-ended and semi-structured, and that all the data from the interviews has been recorded [14].

## 3. Results

All the key informants expressed their satisfaction that the results were relevant to Emirati maternal and infant health in the UAE. The thematic analysis identified and validated five major themes that incorporated all the data recorded from the interviews. These were:Poor communication or miscommunicationRole of the mediaHealth-limiting behavioursThe importance of the education sectorOvercoming barriers to physical activity

### 3.1. Poor Communication or Miscommunication

The participants felt that communication had a significant impact on optimal health outcomes for Emirati women and infants. Communication between health professionals and Emirati patients is important in ensuring that health services are culturally competent. The interviewees recognised this and were particularly concerned about language and cultural barriers in relation to Emirati women:
“Being able to effectively communicate with patients is important. Sometimes this is done through translators, but good translators are hard to find”.
“Health services can be culturally responsive to the needs of Emirati women by first understanding the Emirati culture”.
“UAE has a high number of expatriates working in the health services who are foreigners to Emirati culture; therefore, there is need for them to go through intensive cultural orientations before they engage with the patients”.

These comments highlight that most health professionals in the UAE are foreigners, revealing a need to have more locally trained health professionals to increase cultural competence in the health service provision for Emiratis.

The informants indicated that there is a lack of knowledge regarding the benefits of dietary supplementation, and mistaken beliefs about potentially negative effects of taking supplements. This is compounded by poor communications between Emirati patients and health professionals.
“Iron and folic acid supplements are handed out in plastic bags without any information on the reason that they should be taken during pregnancy”.
“Many women believe that taking any sort of supplements during pregnancy will increase the size of the baby and therefore result in more pain during labour”.

A factor impacting upon Emirati infants’ health is nutrition. The majority of the Emirati women participating in the cohort study breastfed at some stage during the early life of their children. However, complementary foods were often inappropriate or introduced before the recommended four to six months of age [5]. Responses from the key informants suggested that there is a conflict between medical advice, cultural traditions and best practices in relation to complementary feeding:
“I started feeding solids at 4 months and was advised to do this by a nurse at a government clinic”.
“Women need more information on infant feeding as the older women give the younger women advice so bad practices as well as good are passed down through the generations”.
“Chubby babies are given food earlier than skinnier babies as because they are big there is a belief that they need complimentary foods early”.

The growing level of obesity, particularly amongst women in the UAE, is a cause for concern. The cohort study showed that many of the participants were indeed overweight or obese [6]. The causes of obesity are well-documented, but the key informants identified two additional factors which may be of concern in the UAE:
“There is a trend for national women to use drugs to help with weight loss. Many companies are making substantial profits for different types of weight loss programmes, many of which are unhealthy and may even lead to death”.
“During pregnancy, women eat more and are encouraged to eat whatever they feel like as there is a traditional belief that it will negatively affect the baby if the mother doesn’t eat whatever she wants”.

In the area of reproductive health:
“I received no advice on family planning or birth spacing, and have not heard of sudden infant death syndrome”.
“More information on the dangers of marrying relatives is needed as many health problems are passed through close family marriages”.

### 3.2. Influence of the Media on Health

The media in all its forms has developed rapidly in the UAE over the last few years, with a dramatic increase in the availability of local and international television channels and rapid technological advances which have allowed children easy access to the internet. There were both positive and negative influences on the health of Emirati women and infants identified as stemming from the media. One of the key informants expressed concern regarding the negative impact that increased media exposure is having on child health in the UAE:
“Cartoons and other children’s programming include a lot of adverts for fast foods, both of which are now available 24 h a day. This is a recent development as until recently there was limited availability of television and fast food”.

The key informants viewed the mass media as being under-utilised in relation to health content, although essential for conveying health messages to the community. The cohort study highlighted a need for increased health education, and given that women in this population often spend a lot of time in the home, the media has the potential to become a powerful force in disseminating health messages. The key informants identified several suggestions for how the media could be used more effectively to increase health awareness:
“Health information should be communicated through TV channels, magazines and in commercial breaks during local television series”.
“The media play a large role in most people’s lives and currently there are no programmes made locally on health issues”.
“The media should show short informational videos on health topics. These should be culturally appropriate, high quality and not boring or too densely packed with information. The most effective methods of giving health information is through magazines, television programmes and adverts between popular television programmes”.

In addition, the key informants felt that the use of high-profile figures associated with health, such as sports and television stars, could help re-enforce health messages and increase awareness:
“Interest would be heightened if a popular personality such as Carla Mourad (famous nutritionist/television personality based in Lebanon) was used to deliver the health messages”.
“Health benefits can be highlighted on television or in local newspapers and also use of local popular people like women that have and are doing well in sport in the nation as examples for being physically active”.

Participants also recognised the importance of the quality of health information, indicating that it should be presented to women in a variety of ways so that all sectors of the population found it accessible:
“A colourful, informative information pack which provided mothers with a guide of what to expect during pregnancy and the first few months would be useful”.
“Educated people are more likely to collect information from books and leaflets. We should screen health information provided in hospital, and should provide health education through the media, women’s associations, universities and school flyers”.

While the cohort study reported that most of the Emirati female population are well educated, a small proportion of women remain illiterate. Emirati women with lower education levels appeared to be less well-informed regarding health issues. The key informants identified the need to produce health information suitable for women of all education levels:
“Since these are an educated group of women, brochures and TV programmes can be produced that show how breastfeeding can continue and benefits, even if one is working or studying”.
“Health education can be enhanced, particularly for women with lower levels of education, by using more visual aids when providing the information and more demonstrations, for example, when it comes to infant feeding and food preparation. Materials with information on these issues should use culturally appropriate images which are more important than words”.

The informants also acknowledged that the media should be an integral part of multi-organisational health campaigns to increase effectiveness:
“There needs to be joint approach to improve health including educators, health professionals and the media”.

The informants felt that increasing mass media coverage of health issues and improving the quality of information available could significantly improve health education and promotion in the UAE.

### 3.3. Health Limiting Behaviours

Several behaviours which were identified as having a negative impact on Emirati maternal and infant health. A major factor affecting the use of health services is the population’s perception of the effectiveness of these services. The attitudes and beliefs expressed by the key informants relating to the health services may help explain the poor attendance at antenatal and postnatal checkups and clinics reported in the cohort study:
“The doctors have to see a lot of patients and therefore they don’t spend enough time with each woman to explain how to care for herself”.
“Women don’t attend for health checks after pregnancy as there are no set procedures or programmes for post-natal health checks”.
“Many women don’t see the need to attend post-natal appointments unless they have a serious medical problem”.
“The antenatal check-ups for babies are often not well done and often the baby only receives the necessary vaccinations without any developmental checks”.

Two key issues emerge from these statements. Firstly, the health services are perceived as not meeting Emirati women’s needs and as lacking good organisation, and secondly that Emirati women only use them when they are sick and therefore do not understand their role in preventing problems.

The interviews also highlighted that some cultural and religious beliefs have an impact on health related behaviours:
“One of the reasons that I didn’t breastfeed my second child was that it was Ramadan and I felt it was more important to fast”.
“I use car seats for my children, but I know that many people don’t as they believe that it is safer for the baby to be held by an adult”.

Emirati women who are pregnant and breastfeeding often feel conflicted during the holy month of Ramadan. More education and support are needed to enable these women to make decisions based on their own health and the health of their infants at this time.

Some of the chief concerns expressed by the key informants were in relation to children’s diet and levels of physical activity:
“Many children are spoilt and allowed to eat as many sweets, chocolate and fizzy drinks as they like. People think they are being good to their children”.
“Feeding soft drinks, such as Coke, in bottles is commonplace as is the use of French fries and potato chips as suitable foods for children, so children develop poor eating habits from an early age”.
“Schools also sell a lot of unhealthy food and junk food should be removed from schools”.
“Children are not encouraged to exercise or play and stay indoors a lot”.
“Many children suffer from rickets due to inadequate exposure to sunlight”.
“Currently many children are not encouraged to exercise at school and physical education teachers are often overweight and unfit”.

The low levels of exercise continue into adulthood and Emirati women in particular find it difficult to exercise in this society:
“Women feel they don’t have time to exercise as development has meant that many of them go to university and work outside the home as well as looking after their families and running the household”.
“Exercise and physical activity are not encouraged once girls reach puberty and they start to wear the abaya”.
“Families do not encourage women to exercise and they need to be encouraged and supported”.
“Many women have household help and this has resulted in a sedentary lifestyle”.

### 3.4. Important Role for the Education System in Promoting Health

The informants viewed the involvement of the school system as being critical in any measures implemented to improve health education and promotion:
“Schools are an important forum for distributing health information and can be used to educate both parents and children. There is currently very little health education in schools and it is more common in private than government schools”.
“Nutrition and health education should be provided through schools that can influence not just the children but the parents as well. Currently there is very little health education in schools. Health education should be all through the curriculum from nursery upwards. There is a need to develop a complete health education program involving Ministry of Education and Ministry of Health and children, youths and schools”.
“Schools and nurseries also provide an ideal environment for educating women and children about health, especially on topics such as exercise and nutrition”.
“There needs to be a development of health promotion and awareness programme which is coordinated properly, beginning at school age and continuing throughout the lifespan”.

The key informants understood the importance of targeting Emirati children in improving the nation’s health. They also saw the school system as a means of reaching parents to increase their awareness of health issues. The provision of good quality childcare and facilities for mothers to breastfeed and express milk were identified as important factors in enabling women to work and study:
“There needs to be more nurseries with highly qualified staff in workplaces, particularly government workplaces where most Nationals work”.
“There is a need for more workplace nurseries to support breastfeeding”.
“To support mothers who want to breastfeed, there should be more workplace nurseries and facilities for women to express and store breast milk”.
“At workplaces, universities, colleges and the government can make it a mandatory that these places have nursing rooms or day care centres, where mothers bring their babies”.

### 3.5. Overcoming Barriers to Physical Activity

The final theme to emerge from the key informant interviews highlighted the barriers Emirati women face in relation to increasing physical activity levels. As women in the UAE generally do not smoke, drink alcohol or take recreational drugs, obesity and its associated problems are the biggest threat to their health. Although the results from the cohort study did not focus on physical activity, it was evident that physical activity levels were low and the prevalence of being overweight was high. The key informants identified several mechanisms that could be utilised to increase Emirati women’s levels of physical activity. The first mechanism was increased involvement by the whole community:
“Encouraging women to participate in physical activity could be enhanced by having clubs in each community, with good equipment and trainers, encouraging team sports such as volleyball and basketball, aerobics, dancing and swimming”.
“Team sports such as basketball, netball, etc., should be encouraged and fun ways of doing physical activity. Walking is another activity which has decreased drastically as the country has developed. More walkways should be developed and children should be encouraged to walk to school where possible during the cooler months”.
“Women can be encouraged to do more physical activities. If they are working in groups or in pairs, they can motivate each other to go for walks along the beach, for exercise, for a short run, or to the gym”.
“Women could find a way to exercise if they really wanted to. Physical activity needs to be encouraged from an early age”.

These comments suggest that Emirati women working together with friends or in teams would be an important strategy in programmes developed to increase physical activity. A second factor which became apparent from the interviews was the need for exercise and sports to be more accessible to women:
“There should be places for women only where they can enjoy their activities”.
“There are personal trainers available who come to the women’s homes but these are very expensive”.
“Exercising places should be easily accessible, having in mind the culture of Emirati people and also there should be more cost free places for women to do exercises who might not afford it”.

Accessibility in this culture involves having women’s only facilities which are close to residential areas and affordable for Emirati women. There was also a suggestion that there is a lack of awareness of the benefits of exercising:
“There needs to be more education on the risks associated with not exercising and the benefits of improving physical fitness”.

Physical activity in schools is minimal particularly for Emirati girls:
“There needs to be more physical activity in schools currently in many schools. This is virtually non-existent”.

Schools have potential to influence health-related behaviours amongst the Emirati population. Clearly there is a need for health programmes which address the low levels of physical activity in this population.

## 4. Discussion

The key informant interviews authenticated the results from the maternal and infant health study (the cohort study) and provided additional insights into the lives of Emirati women and children, offering a range of positive ideas to improve health. There are multiple interlinked relationships between social context and health for Emirati women and infants in the UAE. This study has highlighted the importance of examining health in a holistic manner, particularly in a developing country such as the UAE where social, cultural and political factors play a significant role in healthcare outcomes and practices.

The cohort study showed that intakes of supplements, such as folic acid and iron, during pregnancy were low, particularly amongst women with lower education levels [13]. This is concerning in a society where there is a relatively high incidence of neural tube defects and maternal anaemia [22,23,24].

Childhood obesity is a concern in the UAE [25] as it impacts on health in later life [26]. Low levels of physical and outdoor activities are already manifesting themselves in the prevalence of childhood obesity, reported as 19.8% in Abu Dhabi [25]. Despite international recommendations, the regulation of advertising in the UAE is lacking and there are frequent advertisements for unhealthy foods on children’s television channels. A lack of regulation on food marketing to children is of concern globally [27], but regulation is strongly recommended. The traditional belief that women should eat what they desire in pregnancy may contribute to high levels of obesity in women in the population. There is a need for national-level programmes to educate Emirati women about nutrition in association with physical activity, a theme covered later. This would also enable Emirati women to make informed decisions about diet programmes before embarking upon them.

It is evident from the key informants that there is a need to improve education regarding the consequences of a poor diet in childhood in Abu Dhabi and throughout the UAE. Recent studies have reported relatively poor dietary choices and dental health in pre-school children in Abu Dhabi [28,29]. The informants also suggest that there is a need for healthcare workers and Emirati families to receive information regarding appropriate complementary feeding for infants.

The cohort study showed that fertility is relatively high and that consanguinity remains prevalent—a practice which can result in serious birth defects [5,22]. The informants’ responses emphasise the need for better communication on pre-marital and family planning counselling services. 

There are many cultural barriers to Emirati women taking part in physical activity as the informants pointed out. These were also reported in a study conducted in the Emirate of Fujairah by Berger and Pearson [30]. One of these factors is the lack of support and encouragement from families, especially after puberty, when women are often forbidden to take part in sports. Another factor is that Emirati women have no time to exercise due to the demands of looking after families or working. Conversely, one respondent suggested that modernisation has reduced physical activity levels as Emirati women no longer carry out household chores.

The informants recognised that childcare standards are often low and Emirati women would feel more comfortable having good quality nurses in their places of work. An important issue is the opportunity to breastfeed. Having workplace nurseries and clean hygienic breastfeeding facilities would enable Emirati women to successfully combine a career and motherhood. The results from the cohort study showed that maternal employment had a negative influence on the longevity of breastfeeding [5].

While the UAE is a wealthy country, many of the issues identified are typical of developing countries, where effective implementation and capacity building lag behind the development of health and social policies [31,32]. For example, while the UAE recognises the importance of primary healthcare, the annual output of 16 family physicians from residency programmes is insufficient to meet the demand now and in the short term [32]. The experience and expertise of Emirati professionals, such as the key informants in this study, provide important insights into the implementation of policies that are largely driven by expatriate advisors and healthcare management companies. The UAE healthcare system is still largely manned by expatriates. [33]. For example, only 1.6% of the 7000 nurses employed by the Abu Dhabi Health Services Company (SEHA) were Emirati [33] in 2016. While there have been considerable advances in many areas of health since the interviews were conducted, the insights from the key informants remain valid. 

## 5. Conclusions

The status of women, cultural beliefs and practices, limited health knowledge, and differences in language used by the local population and healthcare providers and in health promotion materials, all play an important role in determining health outcomes and behaviours. For these reasons, it is essential to consider non-medical determinants of health alongside biomedical determinants to help develop culturally appropriate health strategies for this population. The use of the key informant methodology has highlighted areas for future research and policy, including the communication gaps between healthcare professionals and women, the influences of advertising and the media on health issues, heath education, and ways to increase women’s participation in physical exercise.

### Limitations

The interviews took place in 2007, which in a rapidly developing country such as the UAE means that the results may differ due to changes in areas such as health services, education and the environment in which women and children live. However, no baseline exists to determine whether changes have impacted women and infants’ health and therefore this study provides an important record of this period. The relatively small sample size may limit the ability to generalise from the results. The key informants express their views from their professional positions, and these may not be representative of lay persons’ views.

## Figures and Tables

**Table 1 ijerph-16-03053-t001:** Factors and problems influencing maternal and infant health, on which the interview questions were based.

Factor	Problem	Reference
Folic acid supplements	Women with lower education levels are less likely to take folic acid during pregnancy	[13]
Breastfeeding support for working mothers	Women in employment are less likely to breastfeed for 6 months	[5]
Early introduction of solid foods	Few mothers exclusively breastfeed for 6 months	[5]
Obesity and weight control.	Obesity and its consequences are the most common health problems experienced by women. Physical activity is extremely low amongst the female population.	[6]
Information on sudden infant death syndrome SIDS	Very few women were informed regarding the risk factors which are linked to SIDS	[6]
Safety of infants when travelling by car	Few were placed in a secure car seat when travelling	[7]
Antenatal checks	Attendance at parents’ education and midwives’ classes was low	[13]
Post-natal health checks for mothers	Many women did not have any postnatal health checks after discharge from hospital	[13]
Lack of nutritional knowledge	Women and their cooks and nannies have little knowledge regarding nutrition	[13]
Anaemia	Women experienced high rates of iron deficiency anaemia during pregnancy	[6]

**Table 2 ijerph-16-03053-t002:** Key informants.

Informant	Position
A	Senior community development officer at a local government university
B	Senior director in an organisation providing services for women and children, with an influential role in policy development for these sections of the population.
C	Mother of five, studying health at a local university where she was also a leading member of the student council.
D	Member of Abu Dhabi’s leading family, who has three children and is influential in improving women and children’s health in the emirate.
E	Director of maternal and child health with the UAE Ministry of Health.
F	Paediatrician at the main general hospital in Abu Dhabi servicing the Emirati population
G	Assistant Director of the Centre for Research in Public Health at a local university

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
