# Peer review of "Maternal and Infant Health in Abu Dhabi: Insights from Key Informant Interviews"

_ijerph, 2019, doi:10.3390/ijerph16173053_

Round 1
Reviewer 1 Report
Thank you for your manuscript. I read with interest the study but have a number of questions which needs to be addressed.
1. In the introduction it should be added information about how big problem is overweight / obesity among women and children in Ab-Dhabi (%).
2. It would also be possible to add information about recommendations regarding physical activity and whether women and children meet them (including references).
3. An important problem is also the gestational weight gain (GWG). Are women informed about it? Do they comply with these standards?
4. It is also worth adding information about the influence of parental patterns in nutrition and their influence on the occurrence of overweight / obesity in children.
The above information / data from the literature review can be summarized in a table (similarly to Table 1), which would enrich the merits of the manuscript and make it easier for the reader to understand the subject.
5. Are the answers of key informant free or is it possible to group them?
Author Response
Thank you for your manuscript. I read with interest the study but have a number of questions which needs to be addressed.
Points 1 to 4 all suggest we add general information on the health issues discussed, i.e. obesity, physical activity, gestational weight gain and nutrition. However this information is freely available in published papers, both ours and those of others. These are cited in this manuscript .
Our paper does not aim to provide a detailed report or literature review on these health issues in Abu Dhabi, but instead sought informed and informative views on the results from our previously conducted cohort study. This is stated in lines 58 – 60 as follows:
‘The objective of conducting key informant interviews was to elucidate the results the cohort study of mothers and infants, and to use their insights to formulate recommendations that might improve maternal and infant health outcomes in the Emirate of Abu Dhabi.’
The reader is referred to the published papers reporting the results of our cohort study and also to relevant papers by other authors.
In the introduction it should be added information about how big problem is overweight / obesity among women and children in Ab-Dhabi (%).
The introduction does state ‘Consequent to this rapid change has been a rise in chronic disease; particularly obesity, heart disease and diabetes [1].’ The reference cited is
Hajat, C.; Harrison, O.; Shather, Z., A profile and approach to chronic disease in Abu Dhabi. Global Health 2012, 8, 18.
As obesity was one of the issues identified in our cohort study, it is covered in the published paper detailing that aspect, as listed in Table 1.
Gardner, H.; Green, K.; Gardner, A.; Geddes, D., Postpartum maternal health at a time of rapid societal change in Abu Dhabi, United Arab Emirates. Arab Journal of Nutrition and Exercise 2018, 3, (2), 54-66.
It would also be possible to add information about recommendations regarding physical activity and whether women and children meet them (including references).The reviewer is referred to the general comment above. An additional citation has been added into the discussion:
Berger, G., and A. Peerson. 2009. "Giving young Emirati women a voice: participatory action research on physical activity." Health Place 15 (1):117-24. doi: 10.1016/j.healthplace.2008.03.003.
An important problem is also the gestational weight gain (GWG). Are women informed about it? Do they comply with these standards?
The reviewer is referred to the general comment above. We are not aware of any published work on gestational weight gain in Emirati women in Abu Dhabi. A quotation from a key informant that is relevant is given in lines 159-161.
It is also worth adding information about the influence of parental patterns in nutrition and their influence on the occurrence of overweight / obesity in children.The reviewer is referred to the general comment above.
The above information / data from the literature review can be summarized in a table (similarly to Table 1), which would enrich the merits of the manuscript and make it easier for the reader to understand the subject.
We feel that this is outside the scope of the present paper. Relevant references are cited so that the interested reader can easily locate this information.
Are the answers of key informant free or is it possible to group them?We are sorry, but we do not understand the point that the reviewer is making here. The answers quoted are grouped under the themes identified in the analysis in our paper.
Reviewer 2 Report
General comments: This is a sound piece of research in an area of interest for both developing and developed countries. The small sample size (n=7) and purposive sampling strategy could do with greater discussion/justification regarding their general relevance to the population and ability to make conclusive research statements. It is a well written piece of work.
Minor comments:
Line 35 - Sentence missing its full stop.
Line 39/40 - sentence would be better rearranged.
Line 55 - purposive? not purpose?
Line 76-78 - If the informants were strongly interested/invested can we be sure they had no existing bias/agenda they wanted to express rather than a neutral position on some of the topics?
Table 2 - C Mother of five, G Assistant Director
Line 107 - well suited to (remove as)
Line 164 - add a space away from previous quote.
Line 401-402 - This is probably the most important limitation and needs greater discussion and justification for use of these informants.
Supplement 1 - Stage numbers do not align with the sentences they relate to. Many need full stops at the end. 6 - categorisation or categorization - US or UK spelling needs to be consistent throughout paper.
Author Response
General comments: This is a sound piece of research in an area of interest for both developing and developed countries. The small sample size (n=7) and purposive sampling strategy could do with greater discussion/justification regarding their general relevance to the population and ability to make conclusive research statements. It is a well written piece of work.
The key informants methodology is qualitative rather than quantitative, and hence quantitative sample size considerations are not applicable. The research aims to gather a range of informed opinions and identify themes amongst these. The seven informants in this study are sufficient for this exercise, especially given the relatively few Emirati women in the health services in Abu Dhabi at the time.
Minor comments:
Line 35 - Sentence missing its full stop. DONE
Line 39/40 - sentence would be better rearranged. Rearranged to ‘Due to this influx of migrant workers, more than 63.9% of the population are male.’
Line 55 - purposive? not purpose? Changed to purposive
Line 76-78 - If the informants were strongly interested/invested can we be sure they had no existing bias/agenda they wanted to express rather than a neutral position on some of the topics?
The aim of the key informants interviews was to gather views and opinions on the research findings from the cohort study. Hence the informant’s biases/ agenda are valid opinions. However the methodology, using a thematic content analysis technique, gathers the main converging views of the informants, and they are the results presented.
Table 2 - C Mother of five, G Assistant Director DONE
Line 107 - well suited to (remove as) DONE
Line 164 - add a space away from previous quote. DONE
Line 401-402 - This is probably the most important limitation and needs greater discussion and justification for use of these informants.
This ’limitation’ was added at the request of a previous reviewer, and is worth stating. However the key informants methodology does not seek to document lay persons’ views, but to seek out expert and informed opinions. This requires careful consideration when choosing the informants. We believe this was done and is justified in the text in lines 49-57, 76-78 and 98/99 as follows:
‘A good key informant is seen by Merriam as someone who: “understands the culture but is also able to reflect on it and articulate for the researcher what is going on” [8]. The key informants who took part in this study were all Emirati women with roles in the development of maternal and infant health policy in the emirate of Abu Dhabi. They therefore possessed in depth knowledge of Emirati maternal and infant health, and could offer valuable perspectives. Each had special knowledge, skills or status [9] that they were willing to share to enhance understanding and the interpretation of the cohort study results. Key informants were therefore not selected by random sampling, but through purposive sampling based on their potential contribution to enhancing understanding of Emirati maternal and infant health in the Emirate of Abu Dhabi.
‘All the key informants had a strong interest in maternal and infant health and welcomed the opportunity to contribute to the development of knowledge and recommendations on health issues affecting Emirati women and infants in the UAE.’
‘The key informants were seven Emirati women drawn from a variety of backgrounds, all with involvement in Emirati maternal and infant health services.’
Supplement 1 - Stage numbers do not align with the sentences they relate to. Many need full stops at the end. 6 - categorisation or categorization - US or UK spelling needs to be consistent throughout paper. DONE
Reviewer 3 Report
First of all, its a very old data (from 2007) and definitely major changes have happened since then; though the authors have mentioned that as a limitation. If there is scope, it could be better to compare the situation between 'now' and 'then'. This can come through existing literature search, which is not much visible at the current form of the paper. The small sample size is another important issue, which is also mentioned as one of the major limitations.
The 'Results' section should be more organized and be presented in a more interesting way for the readers. Use of verbatim/quotes are too much. The readers will have to struggle in finding out important information/findings from this section. The authors may select one or two specifically strong quotes under each theme, and present.
The 'Discussion' part should be more linked and well connected with the 'Results' section.
What is the 'take home' message from this study? Should be presented in a stronger way with implications/recommendations.
Author Response
First of all, its a very old data (from 2007) and definitely major changes have happened since then; though the authors have mentioned that as a limitation. If there is scope, it could be better to compare the situation between 'now' and 'then'. This can come through existing literature search, which is not much visible at the current form of the paper. The small sample size is another important issue, which is also mentioned as one of the major limitations.
The reviewer is quite correct that the data is old, but this is clearly indicated in the limitations section, with the justification:
‘The interviews took place in 2007, which in a rapidly developing country such as the UAE means that the results may differ due to changes in areas such as health services, education and the environment in which women and children live. However, no baseline exists to determine whether changes have impacted women and infants health and therefore this study provides important record of this period.’
The aim of our present paper was to report informed and informative views on the results from our previously conducted cohort study. This is stated in lines 58 – 60 as follows:
‘The objective of conducting key informant interviews was to elucidate the results the cohort study of mothers and infants, and to use their insights to formulate recommendations that might improve maternal and infant health outcomes in the Emirate of Abu Dhabi.’
The reader is referred to the published papers reporting the results of our cohort study and also to relevant papers by other authors. There is not much published information on Emirati health in Abu Dhabi, and the available literature is referred to in this paper.
The 'Results' section should be more organized and be presented in a more interesting way for the readers. Use of verbatim/quotes are too much. The readers will have to struggle in finding out important information/findings from this section. The authors may select one or two specifically strong quotes under each theme, and present.
In the reviewer’s opinion, we have included too many quotations. However, in qualitative research such as the present study, these original views are indeed the main reason for publishing the work. We have selected the quotations from the thousands possible carefully, based on the thematic analysis, and included those which we feel make useful points. We do not feel it appropriate to further reduce these quotations as it would detract from the interest and value of the paper.
The 'Discussion' part should be more linked and well connected with the 'Results' section.
The discussion has progressed through numerous drafts based on previous reviewer’s comments. Without specific points from the reviewer, we do not know which areas he is referring to. In our opinion, the discussion links well and is connected with the results. Other reviewers are also apparently satisfied so we would leave this to the judgement of the editors.
What is the 'take home' message from this study? Should be presented in a stronger way with implications/recommendations.
The take home message as provided in the Conclusion and Abstract is the importance of considering non-medical determinants of health alongside biomedical determinants to help develop culturally appropriate health strategies for this population. Further avenues for research as suggested through the key informant interview process are also given.
Round 2
Reviewer 1 Report
The authors responded to the reviewer's comments and clarified unclear issues.
The manuscript can be published.
Author Response
The authors thank you for positive comments. The abstract has been edited to remove extraneous sections and to add more detail on the results.
This manuscript is a resubmission of an earlier submission. The following is a list of the peer review reports and author responses from that submission.
Round 1
Reviewer 1 Report
Thank you for the opportunity to review the manuscript titled “Maternal and Infant Health in Abu Dhabi: Insights from Key Informant Interviews”. Overall, this manuscript clearly described the methodology and results. However, there is still lack of some key information about the background and discussion that the authors should address.
Background:
Authors described in details about the significance of the key informants’ interview methodology and research goal. However, the readers still don’t know what the maternal and child health issues in Abu Dhabi now and what’s the scientific significance to do a qualitative study like this, e.g., is there any research gap for this issue? How this present study will add value to the existing literature? What did the previous work find and anything new for this study?
Method:
Although the authors introduced these seven key informants positions, these interviewees may be well educated with higher socioeconomic status in society (e.g., they have competent English skills, good social/professional career position). I am curious how these women interviewees can represent general women in Aub Dhabi?
Discussion:
I did not see the discussion section in this manuscript, which is a very important section in a scientific public health paper. The authors should consider adding this section and provide a scientific discussion about the findings they found.
Author Response
We would like to thank the anonymous reviewer for insightful comments on our manuscript. We include responses to these comments in the attached document.

Reviewer 2 Report
This is an important topic area, and the title suggests that it has a valuable contribution to make to policy making and scholarship in the area of primary health care provision and public health in middle income countries. However, there are a number of flaws and omissions in the manuscript that require attention before it can be considered suitable for publication.
1. The title does not indicate that the ‘Key Informants’ are in fact UAE policy makers and academics. The title and abstract suggested to me that this study was focused on general population or lay perspectives of women, drawn from a wider population sampling frame to provide their experiences of maternal and child health services in UAE. In fact it is not until the bottom of page two that it becomes clear that the interviews were in fact undertaken with health care / health policy professionals and academics about their impressions of the findings a very old study (2002?). While there is no doubt this is an important group of women to include in this study, given that the study aim claims to about “seeking feedback from Emirati women to clarify the significance of research findings and to formulate recommendations that might improve maternal and infant health outcomes”(line 63) , it is misleading to infer this highly specialised (and most likely high to middle class) group of women can give the sufficient and non-specialised range of key insights that the ‘Emirati women’ label suggests. The authors need to be much clearer and more honest about this earlier in the article, and this includes saying in the title who exactly the ‘Key Informants’ are - i.e. health policy makers and academics.
2. Once it is obvious, which is not the case till later in the paper, that this research is about policy makers' and academics' perspectives and understanding about a population health survey undertaken in 2002, the authors need to make it much clearer which study they are referring to at each point in the article. As it stands, the term ‘study’ using to refer to both interchangeably and this makes the arguments hard to follow. The authors could refer to the studies as Study A and Study B to avoid this confusion.
3. In addition, I wonder if the data for the subject of the paper, is too old? Am I right in thinking that this data was generated in 2007? If so, it seems likely that in the intervening 12 years things will have moved on substantially in the UAE in that time.
4. While the method chosen to undertake the study is appropriate, and I can understand the decision the researcher made to use an existing network of professional colleagues as a focus of this research, I am concerned at the lack of an obvious, alternative (outside the group) perspective that is currently apparent in this research. This fact, coupled with the lack of criticality evident in this piece, causes me to be concerned about the extent to which the researchers were able to stand back and reflect on their personal and professional biases during this work, and the extent to which those coloured the analysis and their interpretation of the data. Where were the exceptional and deviant cases in this work? Everybody appeared to be saying the same thing about each are under investigation; using different words and phrases, but essentially they were saying the same thing. In addition, it is notable that the author contribution list has not been completed in this article, and while reference is made to two colleagues having read the transcripts (page four in table 3), it is not clear who those colleagues where, and importantly, what the various author contributions were.
5. On this the issue of missing information, there are no key words suggestions.
6. The account of the analysis is not complete and presented in a highly unorthodox and unsatisfactory manner from qualitative research, quality perspective way. The table looks like it could have been lifted straight from a text book.
7. The findings are presented as long lists of unattributed quotes (which need attribution of some kind), and this section is currently not in an acceptable format for a qualitative research article. It reads like a list of responses that came straight from the original interview as it took place, and there appears to have been very little attention or energy given to subsequent categorisation, synthesis and interpretation. This section needs significant revision. Far too many quotes have been included, and the author’s need to cite only one (two at the most) illustrative quotes are all that are required, alternatively the authors may choose to tell a more nuanced story of their data in their findings narrative. if they feel the need to include more.
8. In addition, this paper lacks a discussion of the implications of these findings that is situated in the context of other the relevant literature, and the authors need to compare and contrast their findings with relevant literature derived from sources, other than their own previous research. There are a number of interesting issues embedded in the quotes cited, which merit much more discussion and reflection that is currently apparent, not least the position of women living in UAE in relation to their power and agency, in the context of a patriarchal society and social system they live in, to make the changes to their lives and behaviours in the manner the authors (and the key informants) allude to here
9. The voices of women from lower sections of society (whose lives the authors seem most concerned about) and those not working in the public health system, are notably missing from this work, and this is an omission that the authors fail to draw attention to. It is a major weakness in a study that is claiming to be concerned with population health improvement.
10. Whilst I would also agree with the authors’ conclusion that there is a need “to consider non-medical determinants of health alongside biomedical determinants to help develop culturally appropriate health strategies” (line 425 last page), I am not convinced the authors have made it clear what those non-medical determinants are from this study, nor what evidence from their interviews that have to support this claim. The focus of the interviews appeared to be grounded very much in a biomedical paradigm, and therefore this needs much more explanation in the discussion.
11. Finally, what are the additional benefits associated with conducting the interviews ..as it identified other concerns these participants had in relation to Emirati maternal and infant health, which would serve as a useful starting point in the future? (Page 12 last sentence).This needs to be made clearer.
Author Response

(The authors gave the same response as above.)

Round 2
Reviewer 2 Report
1. The text in the abstract and introduction for the most part does make it clearer who the key informants are and this is helpful. However, the abstract should also make it clear - as the introduction now does - that the key informants are in positions within the health care area, including ministries, hospitals, and universities. The current abstract description is still ambiguous i.e Emirati women involved in the maternal and infant health services in Abu Dhabi. At the very least the term ""professionals" should be used - as it is later in the article. In addition, I still contend this should be similarly described more explicitly in the title, as I don't think it is sufficiently informative as it stands, particularly when thinking about researchers undertaking systematic reviews in the future. The key informants in this study are professionals and this should be made apparent in the title.
2. I'm content with the author's response here.
3. I understand the authors' arguments about the challenges of gaining baseline data in this context. However the age of the data according to the journal's guidelines is out of date. So this must be for the editor to decide on.
4. Thematic content analysis does indeed provide a structure for analysis of textual data - but this in itself does not guarantee objectivity or necessarily minimises author bias.
5. I'm content with the author's response here.
6. I'm content with the author's response here.
7. I accept that the small number of informants and the level of detail supplied means that anonymity is a problem in relation to attribution of the quotes .However, I do wonder if the level of detail supplied is in fact appropriate. The individuals described and details of their lives supplied suggest they would be highly recognisable in this context, regardless of attribution. I don’t take issue with the need to provide illustrative quotes of the themes presented. However, it is not clear that all the quotes presented for each theme are indicating or illustrative of their associated key theme. Indeed there are examples where the theme quotes refer to informants’ personal experiences of health care alongside quotes that claim to know the minds and thinking of other women using the services. In the minimum it would be more transparent to see the quotes organised in a way that makes distinction between personal perspectives and experiences and participants’ perspectives about what they believe or have observed about other non-professional women. I still maintain my previously expressed view that this section required significant revision for the reasons I listed previously.
8. I note the additional paragraph in paragraph in section 3. I contend that the discussion is still insufficiently developed. (see comments in 9 and 10 below).
9. The authors appear to have misinterpreted my view about the limitation of their study. I was concerned about the extent to which their informants are expressing views about the mindsets and behaviours about women in general population in their narratives. If those women have been interviewed or surveyed in the same manner as these key informants, as the authors seem to suggest, it is important to that the discussion contains a more direct and explicit comparison and reflection on both sets of findings in order to enable the reader to determine and make a judgement about the extent to which those professional and lay perspectives and reported behaviours converge or not. It is increasingly recognised that professional and lay perspectives need to be understood in their own right, and this issue needs more recognition in this paper.
10. I reiterate the need to summarise and present an explicit account of the key findings and themes within the discussion / conclusion section in line with standard study reporting practice.
11. What issues do the author’s think are a useful starting point for future research?